# How Long Has It Taken for the Physical Landscape to Form? Conceptions of Spanish Pre-Service Teachers

**Alejandro Gómez-Gonçalves** [1,*] , **Diego Corrochano** [2] , **Miguel Ángel Fuertes-Prieto** [2] **and Anne-Marie Ballegeer** [2]

1   Department of Geography, University of Salamanca, 49022 Salamanca, Spain
2   Department of Mathematics Education and Experimental Sciences Education, University of Salamanca, 49022 Salamanca, Spain; dcf@usal.es (D.C.); fuertes@usal.es (M.Á.F.-P.); amballegeer@usal.es (A.-M.B.)
*   Correspondence: algomez@usal.es

**Abstract:** This paper analyses the conceptions of a group of students about geological time and its relation to physical landscape formation, focusing on the frequency and rate of a number of geological processes that have shaped our planet over time and that are involved in the formation of the current relief. Data were collected by means of a questionnaire that was administered to 199 university students from a Spanish public university. A total of 185 of them were pre-service teachers and 14 were geology students. Results demonstrated that pre-service teachers had trouble correctly answering questions about the current relief, but especially those questions related to landscapes throughout the history of Earth. Data analysis was done, taking into account pre-university students' tuition, and results showed that those who had taken a high school branch of sciences and technology obtained better results than those who had taken a humanities and social sciences branch. These latter also obtained better results than those who had taken arts as a high school branch. Furthermore, pre-service teachers had difficulties mastering mid-time magnitudes, which ended up making it difficult for them to understand how the physical landscapes were formed. Finally, from the obtained results, some curricula implications are discussed.

**Keywords:** university students; pre-service teachers; geosciences; geography; deep time; geologic time; landscapes

## 1. Introduction

Landscape is a concept with numerous meanings that is generally used as a hinge element and a transversal axis to articulate concepts of both natural and social sciences. Landscape represents a powerful connection between geography and other Earth sciences, since it allows us to analyse the different layers that make up the environment [1]. Landscape could be defined as a recognizable part of the Earth's surface, with its own physiognomy consisting of a complex network of systems that result from the activity and interplay of rocks, water, air, plants, animals, and humans. A more accurate definition of the landscape is provided by the European Landscape Convention (ELC) [2] (p. 2): "landscape means an area, as perceived by people, whose character is the result of the action and interaction of natural and/or human factors". This is a holistic definition, including all kinds of landscapes, without differentiating between natural or human landscapes. The definition of the ELC has been progressively ratified by different European countries and its acceptance has a number of implications: for example, the convention underlines the importance of the landscape to society as it contributes to human well-being and promotes the connection of cultural identities with their territory [2]. Briefly, the ECL adopts a holistic view that includes four elements of sustainable development: nature, culture, society, and economy [3].

Geologists and physical geographers have a similar definition of landscape and use it for describing geomorphologic features encountered in a certain environment [4]. These natural landscapes are molded by different surficial processes acting over different timescales, from millions of years to days, and thus register the evolution of the surface of the Earth over time. From an educational point of view, in the sense of Lacreu's geological landscape [5], they can be used as an educational resource and as an object of study in itself. According to the author, the adjective "geological" of landscape highlights the elements that can provide information to understand the history of the relief and landscape of an area, even to differentiate between natural and anthropized reliefs.

To analyse the landscape, it is necessary to perform an act of reflection prior to the description. From an educational point of view, a proper analysis of the landscape must include a practical approach that allows hands-on experiences [1]. Geographers have been working on landscape interpretation for decades, and recently some interesting proposals for working with physical landscapes have been published that include both practical and theoretical approaches to understand its evolution [6–8]. In order to transform a resource into an object or problem to be solved by the students, a series of questions on the origin and geological evolution of the landscape could be promoted: Were landforms and their materials always there? How and when were they formed? To answer these questions, it is undoubtedly necessary to have a minimum mastery of the concept of geological or deep time and the great magnitudes of time that it entails.

Physical landscapes are dynamic constructs, in continuous evolution, integrated by a set of elements both visible and non-visible, characterized by being manifest, observable, perceptible. In fact, the above definition of the ELC puts an emphasis on the perceptual dimension of the landscape. The landscape thus can be redefined as "a way of seeing" rather than a scene or an image of the Earth's surface [9]. Landscapes are both materially and perceptually constructed [10]. Perception of the physical environment is influenced by the individual's experiences, and social, educational, and cultural factors. Landscapes can be perceived with several senses, but sight undoubtedly plays a key role. Therefore, it is evident that visual perception depends, among other factors, on the observers and their individual background. However, a proper perception and cognition of physical landscapes cannot be accomplished solely through direct view and individual construction but must be acquired with the help of explanations and exposure to scientific and culturally accepted information. In this sense, it can be suggested that a person who wears certain "scientific glasses" will be able to better perceive, understand, analyse, and assess an observed landscape. That is, where an untrained eye sees mainly the beauty of a scene, a trained person goes a step further, trying to answer how, when, and why such a natural scene has come into being and, most important, will reflect further on the importance of its preservation [11].

The present study stems from the idea that if someone is aware of the time elapsed since the formation and the relief of a certain landscape that he or she is observing, that person will have a better understanding of all the processes involved in landscape formation, as well as the need to preserve it. In fact, landscape practices over an area should start with the determination of geomorphological factors, which always leads to a better understanding and ability to evaluate the entire landscape, its components, and the characteristics of each component [12]. In this sense, some authors stated that [4] (p. 173): "geologists use landscapes as analogues for past environments and relate observed processes in modern landscape (e.g., erosion, or sedimentation) to past processes according to actualistic approaches. Thus, their process-oriented approach requires the development of a four-dimensional view on landscapes looking not only at the surface and the depths below, but also back in time".

Understanding the geologic time (both the relative and absolute time) is fundamental to make sense of the geological factors and processes [13]. Therefore, to properly percept and understand the origin and evolution of natural landscapes, it is necessary to manage the time scales of the different processes involved, both the external ones, such as weathering and erosion, and the internal ones, such as orogenesis. This study explores people's conceptions of geological time and its relation to landscape formation, and more precisely focuses on how Spanish geology students and pre-service

teachers perceive the frequency and rate of a number of geological processes that have shaped the history of our planet and that are involved in the formation of the current relief.

## 1.1. Landscape and Education

The ECL encourages the importance of the landscape in the education system. There is a specific point for "training and education" where the ELC indicates that countries should promote "school and university courses which, in the relevant subject areas, address the values attaching to landscapes and the issues raised by their protection, management and planning" [2] (p. 3). The idea behind the text is to promote landscape education within society in order to increase the knowledge about this issue. Then, this knowledge should qualify citizens to participate, in an adequate way, in making decisions about landscape conservation and landscape management [14].

Specifically, in Spain landscape education is taught in different ways during all educational levels, and different methodologies can be used depending on the characteristics of the students [15–17]. During compulsory education, most of the contents related to landscape are included in social and natural sciences subjects, where students start analysing natural landscapes and then anthropized landscapes, reinforcing the idea of the interdependence between nature and human activities.

Nowadays, environmental education, including natural landscapes, is needed, as most people, especially secondary students, imagine the environment as an idyllic natural landscape where most of its elements are of natural origin, such as mountains, water, or flora [18]. Environmental education should include the importance of landscapes within the context of natural sciences, but also in the context of social sciences, promoting holistic learning that involves the head (knowledge), the heart (emotions), and the hand (skills). In addition, teaching students the basic concepts of landscape formation and evolution will lead to a better understanding of the environmental risks imposed on these landscapes and the need to preserve them, and this will promote public participation in decision-making [19,20].

## 1.2. Geologic Time and Natural Landscape Evolution

Many fundamental natural phenomena occur on extreme time scales that humans cannot experience directly. This is also the case for relief and landscape formation and evolution. But other geological phenomena—for example, volcanic eruptions—occur on a human time scale. In order to understand all these natural phenomena, it is necessary to manage the abstract concept of geologic time (or deep time) and adopt a geologic perspective, taking into consideration that some natural elements have been created over long periods of time under conditions that are impossible to reproduce in a human lifetime scale.

The strong relation between landscape formation and geological time works in both directions, and relief and surficial landscape formation processes provide an excellent framework to understand geologic time and rates as students can imagine how long-term and gradual processes like erosion and deposition shape their surroundings. An appropriate comprehension of geologic time is necessary not only to interpret the relief or the landscape but also to understand key concepts in other areas of geoscience. For example, it is important to link deep time with the dynamic model for geologic change over time because both are the basis of Earth Sciences Literacy [21]. In short, geologic time is a key concept to properly understand the theories of evolution [22,23], plate tectonics theory as a paradigm of modern geology [24], anthropic influence on climate change or the key events in geologic history that include both absolute and relative ages [25,26].

Unfortunately, the ability to think across geologic time scales is difficult for many students, from elementary to undergraduate. The understanding of geologic time requires an ability to deal not only with the large magnitude of time involved but also with the temporal scales of events that occur across deep time [27]. In recent years, in an attempt to identify the main difficulties that students have when managing the concept of geologic time, most research has focused on identifying their conceptions in different contexts and at different educational levels [28]. These studies reveal four main

conclusions: (1) students have trouble identifying and establishing the correct order of key events that have occurred throughout the geological and biological history and tend to mix and overlap some of them; (2) even when the events are spread widely apart from each other in time, students tend to find it difficult to associate them with specific absolute ages; (3) students do not usually relate geologic time elapsed with the appropriate orders of magnitude; and (4) they find it very hard to understand and assimilate the rates and duration of many geologic processes. This last conclusion has played a key role during the design of our investigation as we intend to analyse the ideas on landscape formation of a group of pre-service teachers. Moreover, according to the literature, students tend to explain certain geological processes, especially the ones that occur over very long time scales, such as the origin of a great mountain, with disaster theories such as volcano eruptions, earthquakes, or floods. Difficulties also arise when working with mid-time spans (e.g., formation of valleys, alluvial fans, etc.) [27–29].

It is very important to reflect on how long natural processes take to develop, especially if we think about sustainability, climate change, consumption of resources, loss of biodiversity, or even landscape protection [30,31]. Inclusively, some authors pointed out the importance of mastering geologic time to understand environmental, political, and economic issues related to the sustainability of our planet [32].

In the same research that we mentioned previously, authors investigated the ideas of geology students about deep time [27]. Their results showed that these students, who are supposed to be experts in this field, have difficulties estimating formation rates of landscapes that were formed in intermediate timespans, according to previous results [33]. Another study carried out with college students of geology identified incomplete scientific conceptions about the process of canyon formation and the role of rivers in canyon formation [34]. This is not a new issue as numerous studies that focus on the learning processes of the concept of geologic time have been published in science education [35]. Here we found an interesting reflection because these researches were focused on geology university or pre-university students, and a new reflection arises for us: if geology university students had had problems working with the geologic time, what would happen with those students who studied social sciences or humanities? For example, what is the level of knowledge about landscape formation of future primary or early childhood education teachers, who will have to teach the first notions of landscape and the action and interaction of natural and/or human factors to future generations?

## 2. Purpose

The main purpose of the present study is to investigate the conceptions about how long different landscapes have taken to form in Spanish pre-service teachers, and compare these results with the ideas held by students of the geology degree, here considered as expert students. Thus, this study was guided by the following research questions:

(1)   What are the ideas about physical landscape formation over geologic time held by pre-service teachers and geology students?
(2)   How do pre-university studies influence this knowledge, and more specifically, the type of baccalaureate they have studied?

## 3. Materials and Methods

### 3.1. Design of the Study

This research is based on a quantitative approach and intends to analyse the knowledge of landscape formation over geologic time of a sample of university students. To answer the research questions, a test with 10 multiple-choice questions, adapted from previous published works, was used [34,36]. Non-probabilistic and convenience sampling was employed, consisting of subjects enrolled in the first course of three university degrees [37]. Since data were collected during the first days of class of the first year of the university degree, the obtained results represent the students' pre-university knowledge on the studied topic.

*3.2. Participants and Settings*

Participants of this study were enrolled in the first compulsory university course at three different campuses of the University of Salamanca (Spain). The sample consisted of 199 students: 158 (79.4%) were female and 41 male (20.6%), with an average age of 19.8 years old (range 18–47). A total of 77 students were studying for a bachelor's degree in early childhood education, 108 in primary education, and 14 in geology. The differences in the number of respondents were due to the differences in the number of students enrolled in the first year of the university degrees. We compared the results obtained from students enrolled in education with those obtained by geology students, because the latter are supposed to be the leading specialists in this topic.

All geology university students took scientific pre-university courses (*n* = 14, 100%), whereas education students (*n* = 185) had a more varied pre-university formation: 6 of them studied arts (3.2%), 41 studied sciences (22.2%), 99 studied humanities and social sciences (53.5%) and 35 (18.9%) did not answer this question.

As mentioned earlier, in the Spanish educational system, the concept of landscape and relief appears for the first time in early childhood education, within the unit "interaction with the physical environment". During primary education, the concept of landscape is analysed in social sciences including natural and human factors. The natural components of landscapes are also analysed in natural sciences, where the concept "ecosystem" is studied. In our research, we are going to focus on the most important component of natural factors: physical components of the landscapes. During secondary education, concepts related to landscape appear in subjects related to social sciences and natural sciences. For example, "The terrestrial relief and its evolution" is included in the subject "Biology and Geology" (within the group of natural sciences) and appears in the first and third courses of secondary education. The concept "geologic time", on the other hand, appears as specific curricular content in the fourth year of compulsory secondary education (ages 15–16), as part of the subject "Biology and Geology", but is an elective subject (Royal Decree 1105/2014, of 26 December). After secondary education, the only students who delve deeper into the concept of geologic time are those who take the sciences and technology branch during baccalaureate (pre-university education in the Spanish education system), for example, those students who want to study sciences at the university. Therefore, it is possible that a considerable number of university students have never studied some of these contents during their academic life.

*3.3. Instrument and Data Collection*

The questionnaire used was distributed during the first day of class of the 2019/2020 academic course. This questionnaire (Appendix A) was composed of 10 multiple choice questions with 5 different options: students had to choose the right one, with exception of one question where several options could be selected. Two external researchers from geology and Earth science education contributed to the evaluation of the questionnaire. Four of the questions were selected from the Geoscience Concept Inventory (GCI) v. 3 [36]. The remaining questions were inspired by other researches [27,33,34,38].

All the questions consisted of statements related to the formation of the physical landscape and relief and the frequency and duration of the processes involved. We omitted excessively technical terms from the questionnaire, and no static or catastrophic options were included to answer the long-term and gradual processes analysed. Six of the questions (1–6, see Appendix A) were based on concepts familiar to all students (e.g., mountain formation and erosion, formation of a river valley, etc.) and the other questions explored basic concepts related to the evolution of the landscape throughout the history of the planet (e.g., how long it took Pangea to fully fragment). That is, the questions could be grouped into two major groups: one related to the current relief and the other one to relief and landscapes throughout the history of the Earth.

As mentioned early, the questionnaire was distributed among five groups of students in the course 2019–2020. Two of these groups were primary education degree students from two different campuses of the Faculty of Education of the University of Salamanca: Salamanca and Zamora (Spain).

Another two groups were early childhood education degree students from the same campus and faculty. The last group was composed of geology students who studied at the Faculty of Sciences of the University of Salamanca. The questionnaire was filled out with pencil and paper with no fixed time limit.

### 3.4. Data Analysis

The answers to each question were characterized as shown in Table 1: correct (value admitted by the scientific community), acceptable (erroneous value but not too far from the accepted value), or incorrect (value very far from the one admitted by the scientific community). In a first stage, the data were analysed in terms of frequencies and percentages. In order to understand what factors influenced the investigated conceptions, information was analysed independently between geology and education students. Furthermore, within the education group, the answers were analysed according to the baccalaureate of origin of the students (all the students of the geology degree had studied the science and technology branch).

A more detailed analysis of the collected data required the establishment of a coding system that allowed a statistical quantitative analysis. Table 1 collects the information on which answers were correct, acceptable, or incorrect for each of the questions in the questionnaire. For example, in question 1 (Q1), option 1 is incorrect (1.1 I), option 2 is incorrect (1.2 I), option 3 is acceptable (1.3 A), option 4 is correct (1.4 C), and option 5 is incorrect (1.5 I). The students who answered each question adequately got two points, the acceptable answers one, and the incorrect answers zero. Question 3 (Appendix A), which was the only multiple choice question with two correct answers and no acceptable answers, was coded as follows: students received one point for each item they got right and one point was subtracted for each item they got wrong (although they could never get a negative score). Therefore, the maximum score in the questionnaire was 20 and the worst was 0.

**Table 1.** Answers to the questionnaire and numerical coding.

| Q1 | Q2 | Q3 * | Q4 | Q5 | Q6 | Q7 | Q8 | Q9 | Q10 |
|---|---|---|---|---|---|---|---|---|---|
| 1.1. I | 2.1. I | 3.1. I | 4.1. I | 5.1. C | 6.1.I | 7.1. C | 8.1. I | 9.1. A | 10.1. I |
| 1.2. I | 2.2. C | 3.2. C | 4.2. I | 5.2. I | 6.2. I | 7.2. A | 8.2. I | 9.2. C | 10.2. C |
| 1.3. A | 2.3. I | 3.3. I | 4.3. I | 5.3. I | 6.3. A | 7.3. I | 8.3. I | 9.3. I | 10.3. I |
| 1.4. C | 2.4. I | 3.4. I | 4.4. C | 5.4. I | 6.4. C | 7.4. I | 8.4. C | 9.4. I | 10.4. I |
| 1.5. I | 2.5. I | 3.5. C | 4.5. I | 5.5. I | 6.5. I | 7.5. I | 8.5. A | 9.5. I | 10.5. I |

**Note:** C: correct = 2 points; A: acceptable = 1 point; I: incorrect = 0 point. * In Q3 several options could be selected.

Quantitative analysis was done with the SPSS software. ANOVA (one-way analysis of variance) was carried out, and after it, Tukey's post hoc analyses were performed to identify significant differences (significance level of 0.05) in the means of the groups defined by the university degrees and the pre-university studies. Five groups were considered in the analysis: science students (from the geology degree), education students who studied arts, sciences, humanities and social sciences, and a group of students who did not report their pre-university education (see "Baccalaureate branch" in Table 2).

**Table 2.** Data of mean scores ordered according to the studied high school branch.

| University Degree | Baccalaureate Branch | N | Mean | Std. Deviation | Minimum | Maximum |
|---|---|---|---|---|---|---|
| Geology | Science and Technology | 14 | 13.4 | 2.2 | 10 | 17 |
| Teacher Education | Science and Technology | 41 | 8.6 | 2.5 | 3 | 15 |
| | Humanities and Social Sciences | 99 | 7.4 | 2.7 | 2 | 15 |
| | Arts | 6 | 6.0 | 2.73 | 3 | 9 |
| | No data | 35 | 7.3 | 2.7 | 3 | 14 |
| TOTAL | | 195 | 8.0 | 3.0 | 2 | 17 |

## 4. Results

Figure 1 shows the percentage of answers of 199 students for each question included in the questionnaire (see Appendix A). As can be seen, one of the most remarkable things was that only one question had more than 50% of correct answers (question 6). In addition, four of the questions had 30% of correct answers or less (questions 3, 5, 7, and 10). If we took into account the acceptable answers, percentages of wrong answers noticeably decreased, but only six questions obtained results with less than 50% of wrong answers. These results demonstrated an overall weak knowledge on landscape formation over geologic time, if we consider the complete sample of university students where education students were the majority. Nevertheless, we found remarkable differences between the components of this group, as will be explained later.

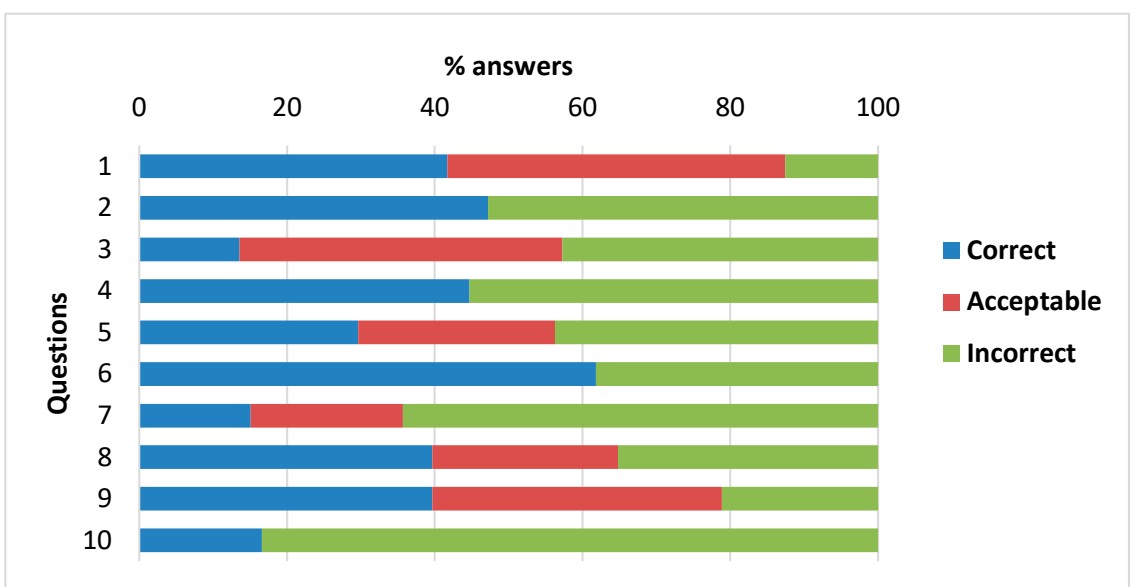

**Figure 1.** Percentage of answers for each question. Note: questions $n^o$ 2, 4, 6, and 10 have no Acceptable answer options.

We differentiated two groups of questions in our questionnaire: on the one hand, we included questions about current relief (questions 1 to 6) and, on the other hand, there were questions about relief and landscapes throughout the history of the Earth (questions 7 to 10). For the first group of questions we registered an average percentage of 39.8% of correct answers, while for the second group we registered only 27.8%. However, if we took into account acceptable answers, the gap between both groups of questions suffered a significant reduction. Question 10 deserves specific attention because this question obtained the highest percentage of wrong answers. It is true that it was one of the questions without the possibility of acceptable answers, but it revealed the difficulty that students have with historical geology questions. In this question, students had to choose among various pictures (see Appendix A) the image that best represented what the planet looked like when humans (hominids) first appeared. So, in order to answer this question correctly, students had to manage, on one hand, the time it took for the tectonic plates to move, and, on the other hand, the chronology of the appearance of the first human being.

In the next step of the analysis we disaggregated the data by university degree. We grouped the results according to students' degrees: one group was composed of students of the early childhood education degree ($n = 77$), another group was composed of primary education degree students ($n = 108$), and the last one of geology degree students ($n = 14$). As we had different sample sizes, in order to know if homogeneity of variances was met by our data, Levene's test was realized before the ANOVA analysis, obtaining a value of 0.202 for the Levene statistic based on mean, with a significance ($p$) of 0.962, and 0.24 based on median, with a significance of 0.944. As the significance in both cases was

greater than 0.05, the null hypothesis of equal population variances was not rejected, and the ANOVA analysis could be performed. We eliminated four students from the general analysis (*n* = 199) due to their peculiar situation: one of them did not complete his baccalaureate, another one did a special baccalaureate branch, and two others took professional training before entering university.

No significant differences were found between primary education and early childhood education students. So, we decided to group these students: in Figure 2 geology students appear on one side, and all education students appear together on the other side. Overall, results showed a great disparity between the two groups of students: On the one hand, in contrast with previous papers [21,26], we observed relatively high scores among geology students and, on the other hand, low scores among education students. Remarkable differences were found between one group and another in terms of knowledge about landscape and relief and the duration of the processes involved: the average percentage of correct answers of geology students to the questionnaire was 64.3%, and the average percentage of correct answers of education students was 32.8%. We expected to find a high success rate among geology students and data collected confirm this assumption, but the low percentage of success rate among education students was remarkable.

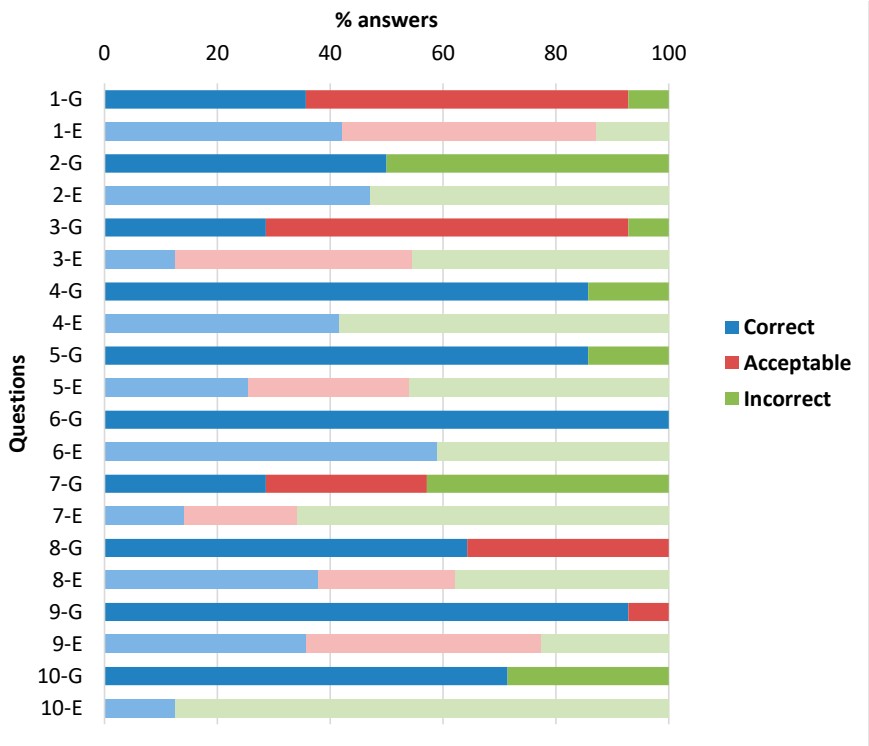

**Figure 2.** Geology (G) students vs. education (E) students: percentage of answers for each question.

If we divided the questions in two groups, as we did in the previous paragraphs, we observed that education students had great difficulties understanding historical geological questions. They obtained a percentage of correct answers of 25%, whereas geology students had 64.3% of correct answers. This big difference suggests that education students have noteworthy difficulties to understand the time it takes for continents to move and to form supercontinents. Again, we have to underline the answers to question 10: the percentage of correct answers of geology students was 71.4% and only 12.4% for education students. This big difference between the two groups of students in the questionnaire reinforced our previously suggested ideas.

When we analysed the questions related to the current relief we observed better results. Education students improved their rate of average correct questions and geology students maintained their scores (37.9% vs. 64.3%). Actually, in question *n*° 1 they obtained more correct answers than

geology students. Education students understood erosive processes relatively well, and these results could be connected with their pre-university tuition. In order to probe deeper, we analysed the results, taking into consideration the branch they studied during their baccalaureate.

Table 2 shows the results of the study ordered according to the baccalaureate program that the students had attended and using the codification system described in the methodology epigraph (a student with 10 correct answers would obtain 20 points; a student with no correct answers would obtain 0 points). Table 2 reveals that students with higher scores (mean = 13.4) were those who were enrolled in geology and took the scientific program during their baccalaureate. On the other hand, students with lower scores (mean = 6.0) were those who studied for education degrees and took the arts baccalaureate program.

One-way ANOVA indicated that there is a statistically significant difference in the mean scores between the different degrees and branches taken during the baccalaureate (F(4190) = 18.218, $p < 0.000$). A Tukey post hoc test revealed that there is a statistically significant difference ($p < 0.000$) in the mean scores between the students who were studying for a bachelor's degree in geology (and took the science and technology high school program) and students who were studying for a bachelor's degree in education and took the high school branches of science, arts, humanities and social sciences, and the group of students who unfortunately did not report their pre-university studies. However, there were no significant differences between the groups of students from the different education degrees. That is, future education teachers who had studied in the science program did not show significant differences in their mean scores compared to students who had studied in the humanities and social sciences or the arts branches. Nevertheless, there were some interesting results. More specifically, we identified noteworthy differences among education students according to their baccalaureate. According to expectations, education students who completed a scientific pre-university formation obtained higher scores than those who completed a humanistic and social formation (39.8% of correct answers vs. 31.3%), and those, in turn, obtained better scores than students who completed an artistic pre-university formation (25% of correct answers) (see Table 3).

**Table 3.** Pre-university formation and percentage of correct and acceptable answers.

| | | Sciences (Geology) | Arts | Sciences (Education) | Humanities and Social Sciences |
|---|---|---|---|---|---|
| | **n=** | 14 | 6 | 41 | 99 |
| Current relief | **1-Correct** | 35.7 | 83.3 | 56.1 | 38.4 |
| | *1-Acceptable* | *57.1* | *0.0* | *36.6* | *47.5* |
| | **2-Correct \*** | 50.0 | 33.3 | 51.2 | 44.4 |
| | **3-Correct** | 28.6 | 16.7 | 17.1 | 12.1 |
| | *3-Acceptable* | *64.3* | *33.3* | *36.6* | *39.4* |
| | **4-Correct \*** | 85.7 | 33.3 | 48.8 | 38.4 |
| | **5-Correct** | 85.7 | 33.3 | 31.7 | 22.2 |
| | *5-Acceptable* | *0.0* | *50.0* | *12.2* | *31.3* |
| | **6-Correct \*** | 100.0 | 16.7 | 68.3 | 59.6 |
| | *Average correct current relief* | *64.3* | *36.1* | *45.5* | *35.9* |
| Historical geology | **7-Correct** | 28.6 | 0.0 | 14.6 | 16.2 |
| | *7-Acceptable* | *28.6* | *0.0* | *24.4* | *21.2* |
| | **8-Correct** | 64.3 | 16.7 | 56.1 | 28.3 |
| | *8-Acceptable* | *35.7* | *50.0* | *19.5* | *26.3* |
| | **9-Correct** | 92.9 | 16.7 | 41.5 | 38.4 |
| | *9-Acceptable* | *7.1* | *33.3* | *39.0* | *36.4* |
| | **10-Correct \*** | 71.4 | 0.0 | 12.2 | 15.2 |
| | *Average correct historical geology* | *64.3* | *8.3* | *31.1* | *24.5* |
| | *Average total correct* | *64.3* | *25.0* | *39.8* | *31.3* |

\* Question with no acceptable option.

One surprising thing was the observed differences among students who studied scientific pre-university courses. Specifically, all geology university students (*n* = 14) completed this high school tuition and also a group of education students (*n* = 41), but when we analysed the percentage of correct answers of these two groups, we identified better rates among geology university students. The existence of optional courses called "Biology and Geology" or "Geology" during their pre-university scientific formation could explain this observation. It is likely that students who were thinking about

undertaking university geologic studies would have chosen the courses related to geology and, on the other hand, those who were thinking about undertaking social university studies, such as primary education, would have chosen other optional courses. No information on these choices was collected in this research but this could play a role in this remarkable difference.

## 5. Discussion and Implications

This study reports the ideas of university students who were studying for bachelors' degrees in early childhood education, in primary education, and in geology about the time involved in landscape and relief formation processes. Since data were collected during the first days of class of the first year of their degree, the results represented the students' pre-university knowledge on the topic. In general terms, geology students had conceptions close to the scientific model, while education students' ideas on the duration and frequency of the processes involved were incorrect. This outcome is surprising because, theoretically, all students who finish their secondary and compulsory studies should have learned the basic geosciences concepts needed to answer questions such as, for example, the timescale of some environmental problems and natural phenomena. In other countries, such as the United States, the National Science Education Standards recommend that at the end of primary education, students should develop an understanding of Earth history and of the fundamental principles of stratigraphy, including a basic understanding of uniformitarianism and catastrophism. Students at this age should also be introduced to scales and rates of Earth processes ranging from seconds (e.g., volcanic explosions) to tens of millions of years (e.g., the erosion of a mountain range) [38,39].

According to our data, baccalaureate branch seems to be a factor that could, at least partially, explain the differences in the knowledge of physical landscape formation processes among the four groups of students. Moreover, information on the subjects that students completed during their baccalaureate could provide even more profound insights into some interesting results, such as the differences observed among students who completed the scientific branch. These differences could be related to the set of subjects they studied. In the Spanish autonomous region where we conducted our study, Castille and Leon, every baccalaureate branch has three types of courses: general core subjects that are common for every single student, specific core branch subjects for each one of the branches (sciences, social sciences, and humanities, arts, etc.) and elective subjects. Among these two last type of subjects, each educational centre offers a different set of subjects, and students choose among these subjects in their baccalaureate. However, this information was not included in this study and a possible relation between the subjects offered in the scientific branch and the knowledge on landscape formation could not be demonstrated. In future investigations it would be interesting to include this information in order to clarify this relationship.

If we consider the baccalaureate studies, geology students who participated in this investigation studied in the technology and science branch during baccalaureate and, thus, probably took several elective subjects related to Earth and environmental sciences. In contrast, many of the early childhood education or primary education degree students took other branches during baccalaureate and, thus, have not worked these subjects of study since secondary education (12–16 years). The question, therefore, is whether the training that students receive during compulsory education about this important concept is sufficient or not to have a general view of how nature works and the duration of the processes that take place in it. As stated before, basic concepts on landscape and its formation are crucial to understand environmental, political, and economic issues related to the sustainability of our planet [32]. Themes of geologic time and rates permeate the Earth Science Literacy Principles, which are a set of big ideas that experts feel citizens should know to make informed decisions on societal challenges requiring geoscientific understanding and solutions.

The vast majority of pre-service teachers studied a humanities and social sciences tuition during their baccalaureate. In the Spanish autonomous region where we carried out our research, Castille and Leon, during this period there was an optional course of geography in the last year. In this course, some units were dedicated to landscape and relief formation, highlighting the importance of geologic

heritage and including the relationship between nature and human beings. So, despite the fact that students who attended the branches of social sciences and humanities obtained worse scores than students who took science and technology, those who did not take the geography optional course during the last year of this period may not have worked on landscape and relief formation since they were 15–16 years old. This fact could play a role in explaining the differences found between pre-service teachers who studied this branch and those who took an arts branch during their high school years.

These results are not surprising if we analyse the Spanish secondary education science curricula in general, and geoscience curricula in particular. In Spain [40]: (1) scientific education receives very little attention in secondary education; (2) geosciences receive less attention than other scientific disciplines; and (3) geoscience contents are usually worked less in the classroom than they should be if taken into consideration of the official programs, and thus are treated in a very superficial and decontextualized way. There is also a lack of didactic and epistemological coherence in curricular designs and school texts due to the fact that the Spanish geology curriculum is outdated and, consequently, it is not capable of generating interest among students and teachers. Because of this, the geology curriculum receives less attention in the classroom, which could finally incentivize the educational administration to reduce its presence even further. Another problem of the Spanish science curriculum in secondary education is that its extension makes it practically impossible to work all the topics contained in it in depth [41]. Finally, teachers are usually poorly trained in sciences contents, and particularly in geoscience contents, and this is reflected in their teaching [41]. In the case of landscape and geologic time, certain previous studies argue that in-service teachers do not feel quite confident when it comes to teaching the concept of geologic time, since they lack the didactic skills and understanding of the subject [42,43].

Results show that, apparently, pre-service teachers have difficulties in mastering mid-time magnitudes. For example, they have significant difficulties in differentiating between the time that tectonic plates take to move around the planet and the appearance of humans (hundreds of millions of years vs. few millions of years). As also noticed in previous studies, these answers also imply that the students do not conceive that humans have been on Earth for a relatively short time, and therefore they resort to the idea of large continents that occurred in the past [28,44]. As a matter of fact, other authors have also demonstrated that one common student belief is that humans first appeared when the Earth had a single continent [45].

The results obtained in this investigation are consistent with other research that also demonstrates that students have difficulties with scales and absolute values [44,46]. In addition, other researches found that university students have difficulties with the ages of the most recent events, like the last glaciation and the appearance of the human being [27,28], something that has also been observed in this study. This lack of knowledge is especially worrying if we consider that we are talking about future teachers, since they will shape children's first ideas about deep time and landscape formation (it seems clear that to teach geologic time and relief formation, teachers will need to have minimal knowledge about this issue). It is true that these students can continue to improve their skills during their university degree, but nowadays, university pre-service teachers' tuition is mainly oriented towards how to teach and, to a much lesser extent, to the contents that need to be taught. To teach the importance of abiotic systems in relief building properly it would be necessary to manage geologic contents and geologic learning strategies and, apparently, students who were enrolled in a bachelor's degree in early childhood education and in primary education do not have the necessary skills.

One of the main problems to solve is that in Spain the vast majority of pre-service teachers took their last course of geology at the age of 15, during secondary education. As mentioned before, the only possibility to review the concepts of the physical landscape and geologic time for students who studied humanities and social sciences in high school was to do an optional course of geography in their last year of high school. So, with the current structure of the Spanish curriculum, geology teachers should highlight some key ideas about relief formation during secondary education, in order to improve scientific literacy and to allow a lifelong learning. Some authors have pointed out that it would be

a good idea to introduce geologic concepts to help students to reconstruct the history of the Earth by studying rocks, their changes over the time, and their distribution in a tridimensional space [47]. But this strategy requires new methodological proposals that go beyond the traditional teaching of conceptual and procedural geological contents. In order to make the contents related to landscape and geologic time attractive for teachers and students, we should find more appealing approaches that motivate teachers and students to work these contents more exhaustively [6]. For example, connecting the consequences of climate change to the possible negative impacts in some of the most beautiful landscapes would allow us to link concepts of geologic time with currently ongoing environmental problems. Other authors propose to restructure the way that Earth sciences are taught during primary and secondary education, including, for example, exercises about the construction of the geologic history of the landscape next to schools [48]. Other authors have demonstrated that students who participated in classroom activities (laboratory activities, small group discussions, written assignments, etc.) had better results in geosciences than those who only read and discussed the units during theoretical lessons [49]. Given the results of our study, we suggest that it could be interesting to reinforce fieldwork sessions during compulsory education in order to consolidate the ideas about the formation of the current relief and, more specifically, those ideas related to historical geology. In fact, previous research has demonstrated that fieldwork has a positive impact on the teaching and learning of surficial processes and their effects on landscape formation [29], as other authors have before when they designed a lesson on Earth history, including a field trip and hands-on activities [50].

Finally, this research presents some limitations. To start with, the sample used in this research is not representative of the entire population of Spanish pre-service teachers. However, this study did allow us to compare our results with other studies that used a similar approach. We could not obtain general conclusions, so we had to interpret the results with prudence. Future work should seek to replicate our findings in other Spanish or European regions in order to corroborate our conclusions. It would have been interesting to have more accurate information about courses that students took during their high school years, in order to elucidate their influence on pre-service teachers' knowledge. Despite these limitations, this research contributes to the existing literature about students' knowledge of physical landscape formation processes, and more specifically, has allowed us to obtain new information about the level of knowledge about geological time that pre-service teachers have.

**Author Contributions:** Conceptualization, A.G.-G. and D.C.; methodology, M.Á.F.-P.; software, M.Á.F.-P.; formal analysis, A.G.-G., D.C. and M.Á.F.-P.; investigation, A.G.-G., D.C., M.Á.F.-P. and A.-M.B.; resources, D.C. and A.-M.B.; data curation, A.G.-G., D.C. and M.Á.F.-P.; writing—original draft preparation, A.G.-G. and D.C.; writing—review and editing, A.-M.B.; visualization, M.Á.F.-P. supervision, A.-M.B.; project administration, D.C.; funding acquisition, D.C. All authors have read and agreed to the published version of the manuscript.

**Funding:** This research was supported by the University of Salamanca under grant MODELGEO-CEI (K133/USAL).

**Conflicts of Interest:** The authors declare no conflict of interest.

## Appendix A

(1) The figure below shows a drawing of a river valley. How long do you think it has taken for the river to be able to excavate the valley?

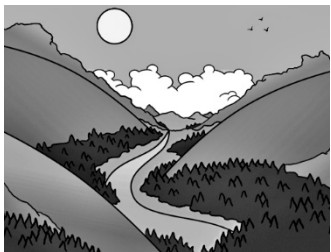

    (a) Tens of years
    (b) Hundreds of years
    (c) Thousands of years ~
    (d) Millions of years *
    (e) Billions of years

(2) The diagrams below represent the profiles of two mountains made of the same type of rock that have finished forming completely in the same place. But which of the following reasons better explains the differences of the two schemes?

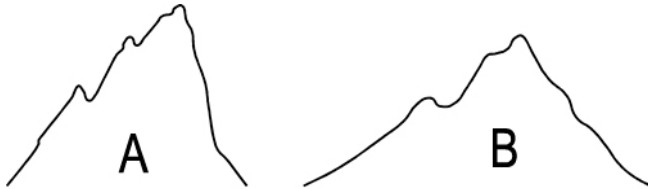

    (a) Mountain A is older than mountain B
    (b) Mountain B is older than mountain A *
    (c) Mountain A is on a continent that moves more slowly than mountain B
    (d) Mountain B is on a continent that moves more slowly than mountain A
    (e) Mountain A has suffered more erosion than mountain B

(3) Which of the following sentences best describes the mountains? (you can check more than one answer)

    (a) The old mountains are higher than the young ones, because they have been growing longer
    (b) Old mountains have smoother slopes than young ones, because they have been eroding longer *
    (c) Old mountains have more vegetation than modern ones, because they have had plants growing on their slopes for longer
    (d) Older mountains have rougher surfaces than younger ones, because they have more time
    (e) All mountains are approximately the same age, regardless of their shape, size, vegetation or roughness *

(4) Do you know how long it takes to form a mountain range?

    (a) Tens of years
    (b) Hundreds of years
    (c) Thousands of years
    (d) Millions of years *
    (e) Billions of years

(5) How long do you think it takes for a mountain to erode completely?

    (a) Tens of years
    (b) Hundreds of years
    (c) Thousands of years ~
    (d) Millions of years *
    (e) Billions of years

(6) When do you think the mountains that we currently see in the landscape will start to erode?

(a)   They are already eroding *
(b)   In tens of years
(c)   In hundreds of years
(d)   In thousands of years
(e)   In millions of years

(7)   And would you know how many great mountain ranges (such as the Himalayas or the Andes) have existed throughout time on Earth?

(a)   Several *
(b)   Tens to hundreds ~
(c)   Hundreds to thousands
(d)   Several thousand
(e)    Millions

(8)   Scientists claim that there was once only one supercontinent on Earth. But how many years do you think it took for the supercontinent to break up and form the continents we see today?

(a)   Tens of years
(b)   Hundreds of years
(c)   Thousands of years
(d)   Millions of years *
(e)   Billions of years ~

(9)   Do you know how many times over the years those great supercontinents have existed on Earth?

(a)   Once ~
(b)   Several times *
(c)   Hundreds of times
(d)   Thousands of times
(e)   Millions of times

(10)  The figure below is a view of our planet from space. The color gray represents land and white represents water. Which of the other figures do you think best represents what the planet looked like when humans (hominids) first appeared?

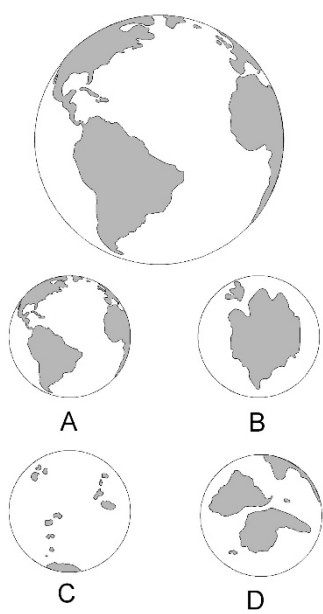



(a)     None

(b)     A *

(c)     B

(d)     C

(e)     D

* Correct answer

~ Minimally acceptable

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
