# Peer review of "How Long Has It Taken for the Physical Landscape to Form? Conceptions of Spanish Pre-Service Teachers"

_education, doi:10.3390/educsci10120373_

Round 1

Reviewer 1 Report

This study evaluates university students’ knowledge regarding geological time and its relationship with landscape formation. Although it handles with relevant issues, this manuscript also presents some aspects that must be improved:

  1. Although I do not feel qualified to judge about the English language and style, I think it needs an overall revision. For example, see lines 218 and 219 of the manuscript.

  1. Regarding the research questions, with this study it is not possible to evaluate the factors that influence the differences between students’ knowledge. We only have access to their pre-university courses. Some assumptions throughout the manuscript were done regarding these factors. But we cannot conclude anything from it (as it is stated in the final section of the manuscript – Discussion and implications).

  1. It does not make sense to compare the group of students that do not report their pre-university education with the other groups. It is not relevant.

  1. Also, I do not understand how it is questioned the ability of teachers to teach geological time and relief formation (lines 417-420). They are at the first year of their university courses. They will improve their knowledge regarding those aspects in the university.

In my perspective, we can only discuss their knowledge at this moment (the beginning of their university course) and how their previous education influenced their knowledge. To evaluate their ability to teach, this study should be conducted at the end of their university course. Also, it could be interesting to compare their knowledge at the beginning and at the end of their university course.

  1. What do you mean by “classroom activities”? Also, the discussion of science curriculum in Spain should be clearer.

Reviewer 2 Report

Dear authors,

Thank you for submitting your manuscript for review, which addresses the knowledge of pre-service teachers around geologic time and landscape formation.
Please find below some suggestions to improve its quality.

First some things to consider about the overall study:
Lines 96-101, Environmental Education is needed to teach not only about the significance of landscapes from a natural sciences view, but also from a social sciences view and to make learning holistic in a way that involves head (knowledge), heart (emotions) and hand (skills).

Please check the accuracy of statement e in your multiple-choice questionnaire in lines 496-497 as it affects the outcomes of your data analysis.

The sample of geology students in low N=14, compared to the sample of education students. In addition, the sample of students who studied arts in their pre-university programmes is very low N=6. Thus, the statistical significance of your analysis may not be important. In this case, you should calculate the effect size to understand better your results. For the ANOVA, did you check first if your data is normally distributed? If not this should be done, and after you may choose the appropriate test.

For your study, you reviewed the secondary school curriculum and some pre-university programmes to check what students are taught during their school/college education. For primary school, you only mention that landscape appears in Primary Education related with social sciences without giving adequate detail on the concepts touched. However, the university students you surveyed will be teaching in Early Childhood and Primary Education and so you should have reviewed those curricula as well. The curricula may not include the concepts of geologic time and landscape formation as you have included them in your survey and thus those teachers may not be required to teach those concepts, but others related with geology/earth science/natural history. This might have influenced the design of your survey to be more targeted toward what those students would have to teach or you could have chosen a different sample of pre-service teachers.

Regarding the limitations of your study, you make a very good point on them and you definitely need to distribute the revised survey to other universities that train pre-service teachers and you should have reviewed what is taught in early childhood and primary education so your insights are more appropriate and your recommendations more targeted.

Some specific observations in the manuscript:
Line 66 please replace he with they
Line 123 please revise sentence as it has grammatical mistakes
Line 147-149 please revise as meaning not clear
Line 166 pretend is not appropriate for this sentence please revise (it is found in other places in the manuscript and needs correction)
Line 176 starts with 77 please revise to Seventy-seven
Line 210 please correct mountain building to mountain formation
Line 313 please revise use of punctuation
Lines 528-535 question is incomplete as b is missing please amend

Reviewer 3 Report

This paper is well structured and presents its results in a comprehensible way. Here are some minor remarks:

The introduction is based on the concept of landscape. But this is not the focus of the research. No question of the questionnaire deals with landscapes but with geological processes and their duration. Also, the connection between knowledge about geological processes in a specific landscape and the need to preserve it (62-64, 66-68) is problematic as no scientific source is cited. Why should knowledge about geological processes reinforces the request for protecting a specific landscape? Therefore, I suggest to point the introduction more to the educational problem of teaching and understanding deep time in school education.

As you mentioned correctly (133-135), students often use disaster theories to explain the origin of geological phenomena. On the other side, you omitted static or catastrophic options when you designed the questionnaire (208). Why? This decision must be explained.

In 246-248 you write “Five groups were considered in the analysis: Science students (from the Geology Degree), education students who studied Arts, Sciences, Humanities and Social Sciences, and a group of students who did not report their pre-university education.” It was not able for me to understand from this sentence, which five groups are meant. Table 2 is a much better oversight of these five groups.

Within the questionnaire the answer b) of question no. 8 is empty.

Reviewer 4 Report

Dear author, below you found my comments and suggestion about your manuscript:

Title

I think that the title can be more specific. For example, the landscape suggest a scientific conception, but in the manuscrit the authors refer the physical landscapes or geological landscapes 

 Abstract

The concept the landscapes formation such as geologic landscapes

Keywords

Add landscapes

 1.Introduction

In the first paragraph, I believe that the authors could be included several references about the landscape:

              Pascual Riesco-Chueca & José Gómez-Zotano (2013) Landscape Fieldwork: Scientific, Educational and Awareness-Raising Requirements in the Context of the European Landscape Convention, Landscape Research, 38:6, 695-706, DOI: 10.1080/01426397.2012.716028

In the second paragraphs, when you spoke about the physical geographers, I believe that you could include various references about the investigations about in landscapes with an important geomorphological componente, such as:

              Beltrán, E. El Paisaje Natural de los Volcanes Históricos de Tenerife; Fundación Canaria Mapfre-Guanarteme: Las Palmas de Gran Canaria, Spain, 2000.

              Beltrán, E. Los paisajes actuales y del pasado de un espacio de montaña volcánica: La Reserva Natural Especial del Chinyero (Tenerife, Islas Canarias). Cuad. Geogr. 2017, 56, 162–186.

              Martínez de Pisón, E. Los paisajes de los geógrafos. Rev. Geogr. 2009, 55, 5–25.

              Martínez de Pisón, E. Saber ver el paisaje. Estud. Geogr. 2010, 71, 395–414. 

1.1. Landscape and education

In this section, I think the some investigations about the teaching landscapes in Spain you could be include, for example in the line 92:

              José Gómez-Zotano & Pascual Riesco-Chueca (2010) Landscape learning and teaching: Innovations in the context of the European Landscape Convention. Proceedings of INTED 2010 Conference. International Technology, Education and Development Conference. Valencia (Spain). 8th-10th March.

              G Alomar-Garau, J Gómez-Zotano, J Arias-García (2017) Teaching landscape in Spanish universities: looking for new approaches in Geography. Journal of Geography in Higher E.ducation. 41(2), 264-282.

1.2. Geologic time and natural landscape evolution

Line 104. For example, volcanic eruptions can occur on human scale.    

Line 133 you wrote is according to the literatura, but in the manuscript you show only one reference, the number 20. I think that you could add some references about this idea.

Line 139 separate the number of references 22 and 23.

Line 151 you wrote twice had

In the last paragraph, you mentioned the reflection about the learning of the teacher about the landscape formation, but my question is the landscape formation over geoogic time (geologic landcapes) or Earth formation? Because landscape has a holistic configuration, and this aspect can fount is in the ELC. 

***

In the introduccion, when you explain the concept of landscapes, I believe that you focused only on their aesthetic appreciation, but the landscape has a configuration that results about the interaction of the natural and cultural factors and elements. Please, you must consider this aspect.

2. Purpose

The question is similar, the landscape formation or geological landscape or Earth formation?

3.Materials and Methods

3.1. Desing of the study

Ok

3.2. Participants and settings

Line 175, Wich universities participated in the study?

Line 186-187 you mentioned "the concept of landscape and relief  appears...",  but I think that you must explain the concept with more detail, because the landscapes associated with the relief forms and processes are different to the concept of geographical landscapes or the landscapes such we found in the ELC.

3.3. Instrument and data collection

Line 206-207, you wrote “All the questions consisted of statements related to the formation of the landscape and relief and the frequency and duration of the processes involved”, Sorry for my insistence, you must explain that the questionsrefer to the landscape over geologic time or geologic landcapes.

Line 207 you wrote “We have omitted excessively technical terms…”, but in the abstract you said “that results show that those who have taken a high school branch of Sciences and Technology have obtained better results than those who had taken a Humanities and Social Sciences branch” I have a question. If you eliminate the technical concepts in the questionnaire, How do you explain the better results of Sciences and Technology students?

Line 215-219, Again, wich universities participated in the study?

3.4. Data analysis

In table 1. i understand the groups of the students (1, 2, 3, 4 and 5) are according to the explain in the lines 215 to 219

4. Results

Line 284 ou mentioned the previous papers, please, add this previous investigations.

Why you don´t use the criteria sex for the analysis of the answers?

5. Discussion and implications

Líne 385-387 you worite “In Spain, during this period, there is an optional course of Geography in the last year. And in this course, some units are dedicated to landscape and relief formation, highlighting the importance of geologic heritage and including the relationship between nature and human beings” This affirmation, which autonomous community in Spain is referred? Because in the 2º Baccalaureate they have a course of Geography ofSpain with some units dedicated to the relief of Spain, but I don't know the whole units in the total autonomous regions of Spain.

Author Contributions

Please add the author contributions

Funding

Indicate that your study haven't received financiation

References

Please I think that you must be add some refereces

Author Response

See the attachement.

Round 2

Reviewer 1 Report

This study evaluates university students’ knowledge regarding geological time and its relationship with landscape formation. Although it handles with relevant issues and all the corrections made, this manuscript still presents some aspects that must be improved:

  1. Regarding the research questions, with this study it is not possible to evaluate the factors that influence the differences between students’ knowledge. We only have access to their pre-university courses. Some assumptions throughout the manuscript were done regarding these factors. But we cannot conclude anything from it (as it is stated in the final section of the manuscript – Discussion and implications). I understand the assumptions done and the comments added, but I think it is not possible to consider research question 2 as a research question, otherwise the questionnaire does not serve its purpose. In fact, the questionnaire was not developed to evaluate those factors, but to evaluate students’ knowledge. I still think the second research question must be improved.

  1. You have eliminated some data regarding the group of students that do not report their pre-university education. Why do you maintain this group of students in table 2?

  1. In table 2, N=195. What happened do the other 4 (as it is stated that the sample consisted of 199 students)?

  1. Please verify the use of “baccalaureate” throughout the manuscript. I think that you are intending to refer to their pre university background.

  1. What do you mean by “There is also a lack of didactic and epistemological coherence in curricular designs and school texts”?

  1. I understand your answer regarding teachers’ education. However, can you better describe their training?
